Optimizing the power of human performed audio surveys for monitoring the endangered Houston toad using automated recording devices

MacLaren Andrew R. amaclaren89@gmail.com 1 2
Crump Paul S. 1 3
Forstner Michael R.J. 1
1 Department of Biology, Texas State University , San Marcos , TX , United States of America
2 Cambrian Environmental , Austin , TX , United States of America
3 Nongame and Rare Species Program, Texas Parks and Wildlife Department , Austin , TX , United States of America
Measey John
Electronic publication date: 2021 Aug 6
Publication date: 2021
Volume: 9
Electronic Location ID: e11935
Received 2021 Jun 11; Accepted 2021 Jul 19
Copyright: ©2021 MacLaren et al.
Copyright year: 2021
Copyright holder: MacLaren et al.
License: This is an open access article distributed under the terms of the Creative Commons Attribution License, which permits unrestricted use, distribution, reproduction and adaptation in any medium and for any purpose provided that it is properly attributed. For attribution, the original author(s), title, publication source (PeerJ) and either DOI or URL of the article must be cited.
License URL: https://creativecommons.org/licenses/by/4.0/

Keywords: Houston Toad, Bufo, Houstonensis, Power analysis, Acoustic, Survey, Automated recording, Detection probability, Presence absence, Amphibian ecology

Funding: The authors received no funding for this work.

==============================
Knowledge regarding the locations of populations of endangered species is a critical part of recovery and facilitates land use planning that avoids unnecessary impacts. Regulatory agencies often support the development of survey guidelines designed to standardize the methods and maximize the probability of detection, thereby avoiding incorrectly concluding a species is absent from a site. Here, using simulations with data collected using automated recording devices (ARDs) we evaluated the efficacy of the existing U.S. Fish and Wildlife Service’s survey requirements for the endangered Houston Toad (Anaxyrus houstonensis). We explored the effect of (1) increasing survey duration, (2) increasing the number of surveys, and (3) combinations of environmental conditions (e.g., temperature, humidity, rainfall) on the detection probability and the number of surveys needed to be 95% confident of absence. We found that increases in both the duration of the survey and the number of surveys conducted decreased the likelihood of incorrectly concluding the species was absent from the site, and that the number of surveys required to be 95% confident greatly exceeded the existing survey requirements. Targeting specific environmental conditions was also an effective way to decrease the number of surveys required but the infrequency in which these conditions occurred might make application difficult in some years. Overall, we suggest that the survey effort necessary to achieve confidence in the absence of Houston Toads at a site is more practically achievable with the use of ARDs, but this may not be suitable in all monitoring scenarios.

Introduction

Monitoring of endangered anuran populations is required to gain an understanding of population dynamics (Pechmann et al., 1991) and the effects of management actions (Walls et al., 2014). Researchers commonly conduct auditory surveys to determine presence or absence of anuran species (Bridges & Dorcas, 2000; Crouch & Paton, 2002; Schmidt, 2003; Pierce & Gutzweiller, 2004; Weir & Mossman, 2005; Jackson et al., 2006). Data from these surveys can also be used for estimates of the relative abundance of calling male anurans (Zimmerman, 1994) or for determining the cadence or phenology of chorusing behavior (Saenz et al., 2006). These surveys are also used to inform land use and development decisions, in addition to ecological research and endangered species management.

There are presently 14 native anurans, with the inclusion of Puerto Rico, classified by the United States Fish and Wildlife Service (USFWS , hereafter) as threatened and endangered (Anaxyrus californicus, Anaxyrus canorus, Anaxyrus baxterii, Anaxyrus houstonensis, Eleutherodactylus cooki, Eleutherodactylus jasper, Eleutherodactylus juanariveroi, Peltophryne lemur, Rana chiricahua, Rana draytonii, Rana muscosa, Rana pretiosa, Rana sierrae, and Rana sevosa). The USFWS has published guidelines for conducting detection/non-detection surveys for only four of these 14 (A. houstonensis, A. californicus, R. chiricahua, and R. draytonii; USFWS, 1999; USFWS, 2005; USFWS, 2006; USFWS, 2007). Two species (R. pretiosa and R. sierrae) have general overviews of currently applied survey methods, authored by the United States Forest Service, but these have not been established as approved policy guidelines. Federal guidelines for conducting surveys of an additional two species (A. canorus and R. muscosa) are reported to be pending approval. The remaining six species lack formal protocols to confirm species presence at a site. These guidelines are intended to ensure that independent researchers are performing standardized surveys designed to maximize the likelihood of detecting the species when present. This lack of survey guidelines, or the existence of poorly designed guidelines, can have serious negative consequences for populations of endangered species. Incorrectly concluding a population of an endangered species is absent from a site can lead to “take” by development of the site through the loss of breeding wetlands and/or upland habitat, as well as potential mortality caused by having individuals present and active in the development site, resulting in preventable losses to the species and expensive fines and protracted delays for the development projects.

Among the existing published guidelines, recommendations for surveying Houston Toads (A. houstonensis) are some of the most specific (USFWS, 2007). These guidelines dictate that at minimum, six 5-minute audio surveys, per year, are required at each listening post (i.e., potential breeding location); surveys must be conducted for three consecutive years; surveys should be spread out between February through April; temperatures must be at or above 57°F; surveys do not begin until about 30 min after sundown and cease if a drop in temperature occurs (presumably below 57°F, however, this is unclear); and wind speeds must not exceed 15 miles per hour (USFWS, 2007). These guidelines also include less quantitative recommendations intended to increase the likelihood that Houston Toads are chorusing on nights that are chosen to survey. These include nights in which humidity is greater than 70%; cloud cover is present or the moon is not full; and rainfall occurring or recent rainfall has occurred. The efficacy of these guidelines has been studied previously; as a result the updated policy now recommends increasing the number of surveys necessary within each season from six to twelve (Jackson et al., 2006; USFWS, 2007).

Many studies have revealed the advantages of automated audio recording systems in determining the influential exogenous environmental factors associated with vocalizing behavior when compared to manual call surveys (Bridges & Dorcas, 2000; Oseen & Wassersug, 2002; Hsu, Kam & Fellers, 2005; Acevedo & Villanueva-Rivera, 2006; Dorcas et al., 2009; Willacy, Mahony & Newell 2015; MacLaren, McCracken & Forstner, 2018b). In general, automated approaches have the capacity to easily collect significantly more survey data over a wider range of environmental conditions and accordingly provide greater statistical power in examining these relationships over manual surveys. Recently, the use of automated monitoring methods has been more formally recommended for Houston Toad detection/non-detection surveys (USFWS, 2020). Thus, the data provided by automated methods provides an excellent opportunity to investigate the true efficacy of manual survey as recommended by the USFWS (2007) guidelines. We are unaware of other studies that model the potential outcomes of a manual survey protocol using data acquired from an automated recording system, but this information is critical to better understand how well the recommended survey guidelines are detecting Houston Toads in particular, and other endangered anurans in general. The use of automated recording methods allows us to investigate the consequences of various important choices in monitoring program design in a systematic and unbiased manner. Thus purpose of this study is to evaluate the efficacy of the existing Houston Toad survey guidelines, (2) to investigate any opportunity to improve manual surveys by increasing the duration of each survey and the total number of surveys conducted each year, and (3) to provide updated recommendations for Houston Toad surveys.

Materials & Methods

Study site

We carried out this study utilizing data gathered from the Griffith League Ranch (GLR), located in Bastrop County, Texas, USA. The GLR is a private property owned and operated by the Boy Scouts of America. This property is commonly represented as the primary recovery site for the Houston Toad (Duarte, Brown & Forstner, 2014), and received population supplementation through captive propagation efforts both during and prior to when this study was conducted. Audio collected from two Houston Toad breeding locations was used. These sites are separated by 2.37 km and are acoustically independent (see MacLaren et al., 2018c). All work conducted to complete this study was done under scientific permit TE-039544-1 issued by the USFWS.

Audio recording and analysis

We used Song Meter SM3 (Wildlife Acoustics, Maynard, MA) audio recording devices to monitor for the call of male Houston Toads at the two breeding locations on the GLR. Song Meters were programmed to record continuously, beginning in January and ending in July, for four years (2015–2018). We powered the Song Meters using rechargeable sealed lead acid batteries (Power Sonic PS-6360 NB, 6V, 36.0 AH). We stored the external batteries in plastic cases, secured to a structure adjacent to each Song Meter. We equipped each Song Meter with four 64GB SD cards for media storage. The additional costs and data storage requirements associated with continuously monitoring limited us to only two locations. We selected monitoring locations based on their history of maintaining a large number of chorusing male Houston Toads relative to other documented Houston Toad chorusing ponds within the GLR.

To analyze the large quantity of audio files collected, we trained an audio classifier using the software Kaleidoscope version 4.3.1 (Wildlife Acoustics). We followed the steps outlined by the manufacturer for completing this process (WildlifeAcoustics, 2017) and used the audio training data provided by MacLaren, McCracken & Forstner (2018a) for the call of the Houston Toad. We chose to simply train towards two “clusters”, Houston Toad vocalizations, and anything that is not a Houston Toad vocalization. This was efficient, and we achieved 100% detection of training vocalizations within a single round of training. However, as discussed by MacLaren, McCracken & Forstner (2018a) focusing on eliminating false negatives (i.e., a Houston Toad vocalization classified in the “not Houston Toad” group) results in a classifier that is permissible to many false positives that must be manually verified as such. This Kaleidoscope cluster was applied to filter all audio recordings for Houston Toad vocalizations. All detections made by the software were manually verified by ARM. We observed during training that detections below 3 s in duration were overwhelmingly false positives, and extremely abundant throughout these data, so we excluded these from Kaleidoscope output prior to manual review. Ultimately, false positive rate for detections greater than 3 s in duration was 0.54, indicating just over half of all detections classified as Houston Toad were not true positives, emphasizing the necessity of manually reviewing all results. We binned detections into 5-minute time intervals, and summarized them as binomial, where 1 and 0 indicate detection and non-detection of Houston Toads, respectively.

Simulation

We simulated Houston Toad audio surveys under three sampling scenarios, each representing a unique approach to subsampling our complete dataset. First, by randomly selecting survey data from the complete pool of recordings from both sites across all years. Second, by restricting available survey data according to environmental conditions presented in the USFWS protocol for conducting surveys for this species (USFWS, 2007). Last, we sought to identify the environmental conditions, if any, that maximize the probability of detecting Houston Toads. All statistical analyses were conducted using Program R (R Development Core Team, 2018). For environmental variables we used the National Oceanic and Atmospheric Administration’s quality controlled local climatic dataset, measured at Giddings, Texas, USA, ca. 25 km East of our sites (WBAN 53979). We utilized moon illumination measured by the US naval observatory (USNO) for Central Time Zones.

The decision to conduct Houston Toad audio surveys is often made in advance and anticipation of appropriate environmental conditions occurring, based largely on weather forecasts. To reflect the uncertainty implicit in this practice we assumed that if environmental thresholds were met at any point within a calendar date, then all data for this date may be surveyed. This is reflected in the results as “dates surveyable” under each scenario. Each of our three survey scenarios selected only for intervals occurring in the months February, March, and April, before 0600 and after 1800 h of each date. This not only reflects roughly what is currently required (USFWS, 2007), it also coincides with peak chorusing activity for the Houston Toad. To implement a random survey selection scenario, all 5-minute intervals within this time frame were considered. To replicate the restrictions within USFWS (2007) we removed dates wherein environmental variables failed to meet the following thresholds: temperature >14 °C, relative humidity >70%, wind speeds <24 kmph, and percent moon illumination <0.5. For our final scenario we searched for alternative environmental thresholds by which researchers may improve success when conducting a human-performed audio survey for the Houston Toad. We calculated summary statistics for the following environmental variables, when Houston Toad vocalizations were detected: temperature (∘C), relative humidity (%), wind speed (kmph), moon illumination (%), hourly precipitation (mm), cumulative precipitation over the previous 24 h (mm), barometric pressure (mmHg at sea level), difference in barometric pressure across 24 h (mmHg at sea level). We then examined which, if any, of these variables offered thresholds that resulted in eliminating the number of dates containing non-detections. We calculated the ratio of detections/non-detections for all combinations of thresholds both above and below all values of temperature, the change in barometric pressure over 24 h, and cumulative precipitation over the previous 24 h. This allowed us to identify which thresholds excluded large periods of inactivity within the breeding season. We then applied these thresholds in the same way as described above and carried out the simulation under these new restrictions.

We removed all instances in which the Song Meters did not record, then pooled all Houston Toad detection data (e.g., 5-min intervals) across the four years and the two sites (N = 92, 652). We randomly selected one 5-minute interval for every date (N = 433) without replacement within the pooled dataset and repeated this 1,000 times. This was done to eliminate the possibility of randomly selecting multiple surveys within the same date, which also more correctly reflects how manual surveys are conducted in practice. We calculated the detection probability as the proportion of positive detections out of the total number of files sampled within each permutation, resulting in 1,000 estimates for detection probability for each simulation. From these 1,000 estimates of detection probability, we calculated the mean and 95% confidence intervals (2.5th and 97.5th quantile) for each distribution of probabilities. We conducted separate simulations for survey scenarios of length 5 min, through 60 min, by increasing 5-minute intervals, resulting in 12 distributions per scenario. We used the formula provided by Pellet & Schmidt (2005) for calculating the minimum number of surveys required to be 95% confident in Houston Toad absence, Nmin= log0.05 log1−p , were p = the simulated estimate of the detection probability for the particular survey scenario.

The detection process consists of many components. First, each site must be occupied during the survey window, then individuals must chorus during the survey window in order for detections to take place. For this study, we assume that for human observers and ARDs this detection process is the same. In practice, human observers may disturb chorusing animals leading to higher rates of false-negative errors, but we did not collect comparable data from human observers in order to test for this potential difference in detection.

Results

We collected detection/non-detection data on 433 dates across all four years (86, 123, 101, and 123 dates for years 2015-2018, respectively). Out of 92,652 5-minute intervals (intervals, hereafter) we detected Houston Toad vocalizations in 3,975 intervals (approximately 4%), occurring among 123 dates (18, 35, 37, and 33 dates for years 2015-2018, respectively). Environmental variables measured for the dates which include these intervals are given in Table 1. Only 122 (∼3%) of intervals containing Houston Toad vocalizations occurred outside of Feb-April. All results henceforth refer to sampling within this peak chorusing period only (72,359 total intervals, 328 total dates, 3,853 intervals with detections).

Table 1 Summary statistics of environmental variables during dates in which Houston Toads (Bufo houstonensis) were detected by automated audio recorders from 2015–2018 in Bastrop County, Texas.

	Min.	1st Qu.	Median	Mean	3rd Qu.	Max.	
Temperature (°C)	−1.15	16.63	19.20	19.24	22.00	31.40	
Relative humidity (%)	14.33	72.33	88.00	82.63	95.67	100.00	
Wind speed (kmph)	0.00	3.67	7.00	7.19	9.67	28.33	
Barometric pressure (mmHg)	29.06	29.38	29.47	29.47	29.57	29.99	
Pressure change (mmHg)	−0.35	−0.11	−0.04	−0.03	0.04	0.38	
Precipitation (mm)	0.00	0.03	0.07	0.24	0.22	4.50	
Moon illumination (%)	0.00	11.00	40.00	48.76	91.00	100.00	

In the first survey scenario, when we selected surveys randomly, the mean detection probability was 0.063 (95% CIs [0.038–0.100]) for surveys 5 min in duration, and 0.121 (95% CIs [0.088–0.171]) for surveys 60 min in duration (Fig. 1, Table 2). These probabilities result in requiring a mean of 47 (range = 29–79) surveys to be 95% confident in determining absence of the species when conducting 5-minute-long surveys, and on average 24 (range = 16–33) when conducting 60-minute-long surveys (Fig. 1, Table 2).

Figure 1 Probability of detecting Houston Toads, and the number of surveys needed to infer absence, as duration of survey increases.

Results of simulation to assess mean probability of detection of Houston Toads (Bufo houstonensis) (bottom panel), and the mean number of surveys necessary for a given probability of detection (top panel), and their 95% confidence bounds, as the length in minutes of each auditory survey increases along the x-axis, for three approaches to survey selection: Random selection (left), following USFWS, 2007 (middle), and under our proposed optimization for survey selection (right).

Table 2 Mean and 95% confidence bounds for the probability of detection and number of surveys required to be 95% confident in absence of Houston Toads (Bufo houstonensis) during acoustic surveys, for three sampling paradigms, as duration of survey increases from 5 to 60 min.

	Detection probability	Number of surveys	
Duration (mins)	2.50%	Mean	97.50%	2.50%	Mean	97.50%	
	Randomly selected	
5	0.038	0.063	0.100	79	47	29	
10	0.046	0.074	0.113	64	39	26	
15	0.050	0.082	0.121	59	35	24	
20	0.058	0.089	0.129	50	33	22	
25	0.063	0.094	0.138	47	31	21	
30	0.067	0.099	0.142	44	29	20	
35	0.071	0.104	0.146	41	28	19	
40	0.075	0.107	0.150	39	27	19	
45	0.075	0.111	0.154	39	26	18	
50	0.079	0.115	0.158	37	25	18	
55	0.083	0.118	0.167	35	24	17	
60	0.088	0.121	0.171	33	24	16	
 	USFWS protocol	
5	0.042	0.080	0.119	70	37	24	
10	0.051	0.094	0.136	58	31	21	
15	0.059	0.103	0.153	49	28	19	
20	0.068	0.110	0.153	43	26	19	
25	0.076	0.116	0.161	38	25	18	
30	0.076	0.121	0.169	38	24	17	
35	0.085	0.126	0.169	34	23	17	
40	0.085	0.129	0.178	34	22	16	
45	0.085	0.133	0.178	34	21	16	
50	0.093	0.136	0.178	31	21	16	
55	0.093	0.140	0.186	31	20	15	
60	0.093	0.143	0.186	31	20	15	
 	Optimized protocol	
5	0.060	0.101	0.143	49	28	20	
10	0.075	0.118	0.165	39	24	17	
15	0.083	0.128	0.173	35	22	16	
20	0.090	0.136	0.180	32	21	16	
25	0.098	0.143	0.188	30	20	15	
30	0.105	0.149	0.195	27	19	14	
35	0.113	0.154	0.195	25	18	14	
40	0.113	0.159	0.203	25	18	14	
45	0.120	0.163	0.203	24	17	14	
50	0.120	0.167	0.211	24	17	13	
55	0.128	0.170	0.211	22	17	13	
60	0.128	0.173	0.218	22	16	13	

Our second survey scenario, in which we removed all dates that did not reflect the environmental thresholds given by USFWS (2007), resulted in reducing the number of available survey dates to 118 out of 328. Under this scenario only 1,737 intervals containing detections are available to be sampled, leaving 1,895 intervals (49%) known to possess Houston Toad vocalizations unobservable to surveyors (3,853 intervals total). The mean detection probability was 0.08 (95% CIs [0.042–0.119]) for surveys 5 min in duration, and 0.142 (95% CIs 0.093 –0.186) for surveys 60 min in duration (Fig. 1, Table 2). These probabilities result in requiring a mean of 36 (range = 24–70) surveys to be 95% confident in determining absence of the species when conducting 5-minute-long surveys, and on average 20 (range = 15–31) when conducting 60-minute-long surveys (Fig. 1, Table 2).

In our final scenario, calculating the proportion of detections to non-detections over a range of environmental thresholds revealed that a unique combination of temperature (>16 ° C), precipitation (>0 mm/day), and change in barometric pressure (<−0.07 mmHg) provided the greatest advantage to surveyors. These thresholds allow 133 dates to be surveyable, comparable to USFWS (2007), yet provide 2,569 intervals containing detections. Under this scenario, the mean detection probability was 0.105 (95% CIs [0.066–0.146]) for surveys 5 min in duration, and 0.179 (95% CIs [0.133–0.229]) for surveys 60 min in duration (Fig. 1, Table 2). These probabilities result in requiring a mean of 28 (range = 20–49) surveys to be 95% confident in determining absence of the species when conducting 5-minute-long surveys, and a mean of 16 (range = 13–22) 60-minute-long surveys to be 95% confident in determining absence of the species (Fig. 1, Table 2).

Discussion

This study demonstrates that the existing USFWS (2007) guidelines for conducting human performed surveys result in a likelihood of concluding absence of Houston Toads from a site when they may be truly present. Given the endangered status of the species, and the precedent set by Jackson et al. (2006), researchers should attempt to achieve 95% confidence in conclusions of absence. Currently, 36 (12 per year, for 3 years) 5-minute-long surveys (180 min total) are required to determine absence (USFWS, 2007). However, our simulation reveals that up to 79 surveys (395 total minutes) of this duration may be required to adequately determine Houston Toad occurrence at a single site. Failing to detect Houston Toads when they are truly present ultimately results in undetected populations, which in the event that the monitoring is being performed as part of a development project, may contribute to local extirpation events. This is especially true for populations outside of our study site, which do not receive any form of population supplementation, and may only support a few individuals. It is also important to add that our study, and the study conducted by Jackson et al. (2006), were carried out using data collected from locations where Houston Toads are likely at, or near, their highest abundance, and as such the data presented here could underestimate of the survey requirements necessary to detect other, smaller populations (Tanadini & Schmidt, 2011).

Our approach demonstrates that the false negative errors associated with manual surveys can be reduced in three main ways. First, manual surveys of longer duration can be performed. In each of our scenarios, the detection probability increases as survey’s duration lengthens. Second, the number of surveys performed can be increased. Third, surveyors can target occasions in which environmental conditions are most closely associated with chorusing behavior among Houston Toads. The trade-offs associated with each of these methods of decreasing false negative errors will depend on the timeframe and scope of the specific Houston Toad monitoring project. Our results indicate human observers must, at minimum, more than double the amount of time spent surveying in order to determine absence with 95% confidence. However, for human observers these changes may be too onerous to allow manual methods to be feasible for determining detection/non-detection from a site. An alternative approach would be the use of automated recording devices (ARDs), as they are demonstrably effective at the task of detecting anurans, especially when rare, due to the efficiency with which high cumulative detection probabilities are achieved (Hsu, Kam & Fellers, 2005; Acevedo & Villanueva-Rivera, 2006). Automated audio recording devices designed for monitoring wildlife are becoming smaller, more affordable, and are able to collect audio much more frequently than human observers (Saenz et al., 2006; Aide et al., 2013, Willacy, Mahony & Newell 2015).

The goals of this study, and that of the federal protocol we evaluate (USFWS, 2007; USFWS, 2020) are limited to determining occurrence of the Houston Toad at a single site (i.e., one human listening post or ARD location). It is our view that human observers should only be employed towards site specific detection of Houston Toads when automated methods are not suitable or potential recorder placement is not permitted. This is particularly relevant for the Houston Toad as the majority of Texas is privately owned lands and public roadways enable access across remaining habitat patches. Further, human observers might better be employed conducting survey methods that cannot adequately be conducted using remote, passive, methods. For example, in order to detect Houston Toads that do not chorus (i.e., subadults, females) human observers may be required to employ drift fence arrays (Brown, Swannack & Forstner, 2013) or sample aquatic habitats. Additionally, methods of determining anuran occurrence at larger scales (i.e., county or regional) often require visiting many sites in a single day, and have successfully been implemented using human observers (Gorman, 2009). Yantis & Price (1993) employed similar methods to determine the distribution of the Houston Toad within Texas.

One critical aspect in the application of our findings is the frequency with which survey dates containing suitable environmental conditions can be expected to occur. If we consider the four years utilized in this study (2015–2018), we find that conditions permissible under the currently accepted USFWS (2007) guidelines only occur on 27, 39, 39, and 31 dates within peak chorusing period each year, respectively. Our simulation indicates that up to 79 5-minute-long surveys should be conducted to minimize false negative errors, and accordingly surveyors would be required to choose an alternative approach (i.e., surveys of longer duration) in order to achieve confidence in their findings. If we consider the environmental conditions discussed in our third scenario, we find that for years 2015–2018, these conditions only occur on 40, 53, 5, and 50 dates within the peak chorusing period each year, respectively. While these results illustrate that in certain years this increases the abundance of surveyable days, during 2017 this method provides only five days of survey appropriate conditions, which reduces the practicality of this approach considerably. Additionally, it would be unwise for researchers to restrict their survey design to a limited set of environmental conditions when these events occur stochastically, and are not guaranteed to occur the necessary number of times to have confidence in conclusions of absence. A caveat within this scenario is that the range of conditions we identified during are derived from our complete dataset, and have not been applied to an independent group of data, and so our results from this scenario may be biased optimistically.

Given the environmental and temporal constraints we have identified, researchers have much to consider when scheduling acoustic surveys for the Houston Toad. The cost to researchers performing acoustic surveys using human observers is likely variable, and we will not speculate on the individual cost of each survey. Although ARDs are, in the long term, less expensive than human performed surveys, factors such as travel and wages can be minimized (Williams, O’Donnell & Armstrong, 2018). We believe our approach demonstrating detection probability as it increases with survey duration provides researchers under variable circumstances options to achieve confidence in absence determinations. For example, if travel costs are large (i.e., researchers must travel a great distance to reach their field sites) perhaps fewer surveys of greater duration may be favored, whereas researchers with many sites to visit may prefer shorter surveys, allowing more sites to be visited on each calendar date.

Conclusions

Given the findings of our simulations, we strongly recommend that human observers restrict their surveys to the peak of Houston toad activity that occurs during the 89 day period between February 1 and April 30. While we believe it is wise to use a priori knowledge of the environmental conditions in which chorusing generally takes place to improve the likelihood of detecting Houston Toads (MacLaren, McCracken & Forstner, 2018b), our study reveals that these events are rare, do not consistently elicit vocalization behavior, and may not allow for adequate effort to be put forth by human observers in any given year. For these reasons we feel our first scenario is most applicable, in which surveyors can choose to survey any date. Houston Toads are in decline throughout their native range (Forstner & Dixon, 2010) thus, we believe that, due to the serious consequences of potential false negative errors, the upper 95% confidence interval for randomly selected surveys be adopted as the minimum survey effort threshold (Table 2). Situations that trigger the need to conduct surveys following (USFWS, 2007; USFWS, 2020) are likely to occur in areas where local occurrence is not known (e.g., marginal habitats). Marginal populations are in the most need of stewardship, and maximum survey effort (i.e., the upper bounds of our confidence intervals, or beyond) is likely necessary to detect these less abundant population remnants.

We found that previously suggested environmental correlates to chorusing among Houston Toads offered improved detection probabilities over randomly selected surveys. However, we found that not all suggested weather criteria within USFWS (2007) were useful, specifically, moon illumination, humidity, and wind speed. This is either because these variables share no true relationship with chorusing within Houston Toads, as is the case for moon illumination, or because they do not serve as a hard boundary, as is the case for relative humidity. For example, relative humidity may range between 10% and 90% within a given single date in response to natural diel cycle. We identified definitive thresholds among temperature, precipitation, and shifts in barometric pressure that improve the probability of detection for Houston Toads beyond what USFWS (2007) currently suggests.

This study updates and expands upon the findings of Jackson et al. (2006). For perspective this previous study (Jackson et al., 2006) utilized twenty 5-minute surveys (100 min) within a given year at a single site, whereas within a single year one ARD provided us with approximately 60,000 min of audio from a single site. Using these vast and detailed data we found that detection probabilities, for surveys of any length, and under any sampling scenario, were lower than what has been previously estimated for this species (Jackson et al., 2006). By suggesting more accurate environmental thresholds under which surveys should be conducted, and evaluating surveys of varying duration, we have provided researchers and managers with an approach that should make the highest probability of detecting Houston Toads possible. Our approach to simulating survey effort allows researchers to choose the combination of survey duration and number of surveys they find most appropriate and maintain 95% confidence in determinations of absence. Like Jackson et al. (2006) our results suggest that the USFWS should modify the mandatory survey guidelines to require more surveys in each season than is currently specified. Moreover, for surveys that are designed to determine presence or absence towards regulatory decisions at a site, the conflict between the availability of suitable environmental conditions and the importance of conducting sufficient surveys based on environmental factors that increase the probability of survey success, ARDs should be strongly considered where possible. Finally, it is critical to differentiate absence determinations made from chorusing data from true absence of this endangered anuran from a potential disturbance site given the underlying nature of juvenile amphibian dispersal and adult use of upland habitats.

Supplemental Information

Supplemental Information 1 Occurrence data for the Houston Toad

Click here for additional data file.

Supplemental Information 2 Environmental data corresponding to occurrence data

Click here for additional data file.

Supplemental Information 3 Simulation R code

Click here for additional data file.

We thank the Capitol Area Council-Boy Scouts of America for their cooperation, and Shawn McCracken, Floyd Weckerly, Joe Veech, Andy Royle, Ben Bolker, and Charles Hermann for reviewing earlier drafts of this work. Additionally, we would like to thank Brian Halstead and Res Altwegg for serving as reviewers, their thoughtful contributions and suggestions are appreciated.

Additional Information and Declarations

Competing Interests

Author Contributions

Data Availability

The authors declare there are no competing interests.

Andrew R. MacLaren conceived and designed the experiments, performed the experiments, analyzed the data, prepared figures and/or tables, authored or reviewed drafts of the paper, and approved the final draft.

Paul S. Crump and Michael R.J. Forstner conceived and designed the experiments, authored or reviewed drafts of the paper, and approved the final draft.

The following information was supplied regarding data availability:

Occurrence data (Houston Toad presence-absence) and environmental (weather) data and the code (R script) is available in the Supplemental Files.

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
