# Peer review of "Optimizing the power of human performed audio surveys for monitoring the endangered Houston toad using automated recording devices"

_PeerJ, doi:10.7717/peerj.11935_

## Round 0.1 · original submission · Minor Revisions

Thanks for your submission to PeerJ. The reviewers have now assessed your manuscript, and find that it will be suitable for publication after some minor revisions. I concur with their assessment and invite you to address their comments accordingly.

·

Basic reporting

There are a few clarifications needed (see 'General comments' section). The raw data are shared along with R code. I suggest that you provide the code that actually reads your data set in to R, with all the correct formats (separators, etc), instead of using '=read.csv(file.choose())'

Experimental design

no further comment

Validity of the findings

no further comment

Additional comments

This study examines detection probabilities of Houston toads at two localities where they are known to occur. The study makes use of data collected continuously using acoustic recording devices (ARDs) and then examines the rate at which calls were detected in 5-min intervals. This is used to evaluate the effectiveness of the guidelines for conducting presence / absence surveys for this species. The main results are that the current guidelines lead to rates of false absences that are far higher than 5%.

One general thought I had is that you had to assume that the detection process of ARDs followed by a classifier leads to the same non-detection rates as human observers would. I don’t think this is a big problem but should be treated a bit more explicitly. I think it would be helpful to acknowledge that the detection process consists of several components. If a site is occupied, the species may not be detected because the frogs are not in the pond at the time (not sure whether this applies to the Houston toad). If they are in the pond, they may not call at the time of the survey. If they are there and call, the observer / ARD may fail to record them. Which of these components are relevant for your study (from reading your ms I had the impression that the toads don’t always call but may be wrong in this)? And how similar are those components of the detection probabilities the same for ARDs and humans?

Line 53: do the guidelines specify what the goal of the surveys should be? 95% probability of detecting the species at a site where it occurs? Or is that not clearly specified?

Line 139 and following: clearly define what you mean by ‘sampling protocol’ and ‘survey scenario’ and then use these terms consistently. E.g. in line 187, you use ‘sampling protocol’ for the different survey durations, which is different from line 140.

Lines 186 – 188: explain in a bit more detail how you calculate the confidence intervals. Did you calculate the confidence interval for the detection probabilities as the 2.5th and 97.5th percentile from the 1000 repeat simulations? And then plug those into the equation in line 185?

Line 213: “average 17 (range = 15-30)”. Are these numbers correct? In all other cases, the average was closer to the middle of the interval.

One issue with the third survey scenario (where you explore more suitable environmental conditions) is that you test the method on the same data you used to develop it. This needs to be stated as a caveat and the results treated appropriately, i.e. as a proposed strategy that needs to be tested on new data. Your results are probably too optimistic.

Another caveat, and you mention this one, is that your study included just two sites that may be unusual in terms of density of the species. I think this is really important since abundance is often a key driver of detection probability. In this connection, you could also consider how your results relate to the fact that optimal survey effort depends on the goal of the assessment and if the goal is to establish absence with a certain probability, one also needs estimates of occupancy (Guillera-Arroita, G., J. J. Lahoz-Monfort, M. A. Mccarthy, and B. A. Wintle. 2015. Threatened species impact assessments: Survey effort requirements based on criteria for cumulative impacts. Diversity and Distributions 21:620–630).

Line 306: “...found that not all suggested weather criteria within USFWS (2007) were useful...” did you? You don’t describe how you analysed the effectiveness of individual criteria, although this could easily be done, for example by using logistic regression.

One main take-away message I got from this study is that one should really deploy ARDs if it is important to establish absence, and this is also a point you are making. I think you have a great opportunity to go a step further and explore how many days these devices would have to be left on site to establish absence with 95% and 99% (say) probability. I think that would make your argument much stronger and help towards establishing guidelines for using ARDs in such situations.

The information in Table 2 and Figure 1 is the same. Both formats are probably useful but the table could be given as an appendix.

·

Basic reporting

Overall, the basic reporting in this manuscript was good. The Introduction was structured well—it introduced the problem, placed the importance of the study in context, and concluded with the three objectives of this study. The Methods were clear in most places, though I found some aspects of the description of the simulation to be confusing (as outlined below in the Experimental Design section). I found the Results clearly presented and easy to understand. The tables and figure supported the text nicely, but in Table 1 it would be interesting to present the same summary statistics for dates when Houston Toads were not detected to provide a contrast that might indicate how limiting environmental conditions are with regard to toad chorusing. Also, in Table 2 and in the text of the Results, it might be best to just round up to the next integer for presenting the number of surveys required to be 95% confident in absence because surveyors are unlikely to do a fraction of a survey.

Some minor grammatical errors and suggestions are indicated in the attached commented pdf.

Experimental design

The manuscript presents original primary research within the scope of PeerJ. The research questions are well defined and relevant to conservation of species that vocalize, and they are presented clearly at the end of the Introduction. The Introduction further provides information about the knowledge gap filled by this research. The investigation was rigorous and performed to high ethical and technical standards. For the most part, the Methods were described with enough detail to replicate them, but I thought that some additional detail and clarity about the automated audio classifier would be helpful.

On lines 125–137, the authors do a nice job by providing a good amount of detail about the Kaleidoscope automated audio classifier’s performance. The sensitivity of the classifier was impressive (lines 130–131), and the authors provide information indicating that the calls < 3 s duration identified by the classifier were overwhelmingly false positives (lines 133–135). I was left wondering, however, about the false positive rate was for putative calls > 3 s in duration. This information would be useful for others seeking to use automated classifiers in their research.

Validity of the findings

The authors provide the underlying data in the supplemental files. Because this is an observational study, there are not controls per se, but the authors provide information on how the data were screened and vetted for quality control in the manuscript. The one exception, outlined above, is that I think additional information about false positives for calls > 3 s duration would be valuable information to present. The authors presented little speculation and their conclusions were well-stated and linked to the original research questions. Importantly, the authors addressed the potential that by selecting sites with high abundance for their study, detection probabilities could be biased high, and they caution readers that finding smaller populations will likely require additional effort (lines 239–241 and 301–303). I have two main comments that I think would improve the Discussion:

1) On line 229 in the statement, “…result in a likelihood of false negative errors that is too high…,” I find “too high” to be so vague as to be of little use. What is an acceptable rate of false negative errors? Is it anything > 5%, as suggested by the focus on declaring Houston Toads absent with 95% certainty, or some other threshold? Please clarify. I found the statements “an increase is required” and “to adequately detect” on lines 249 and 250, respectively, similarly vague.

2) In general, I think it would be beneficial to provide some further guidelines on survey design. Specifically, on lines 250–252, I think it would be useful to illustrate the total effort necessary for human observers using manual methods. For example, based on Table 2 (and rounding the number of surveys up to the next integer), under the optimized protocol 28 (20–49) 5-min surveys require 140 (100–245) min of total survey time, whereas 16 (13–22) 60-min surveys require 960 (780–1,320) min of total survey time. Aside from using ARDs, what survey duration would you recommend for future surveys? How do the findings outlined in the paragraph on lines 273–285 (number of days meeting survey criteria per year) factor into this recommendation? Is the cost of surveys a consideration, and if so, how does costs scale with survey duration and the number of surveys?

Additional comments

This manuscript addresses an important topic for conservation: When have we looked hard enough to declare a species absent from a site? The authors use detailed, continuously collected data from ARDs and a simulation study to evaluate three different scenarios for conducting manual auditory surveys for the endangered Houston Toad. Importantly, the authors identify environmental thresholds that improve the likelihood of detecting chorusing Houston Toads. Unfortunately, these conditions can be so limiting in some years that one can have little confidence that Houston Toads are absent from a site at which they are undetected. Overall, the manuscript was a joy to read and addressed an important and interesting topic.

---

## Round 0.2 · accepted · Accept

Thank you for your revision.

I would like you to clarify the taxonomic status for this species. Your manuscript (L50) appears to accept that some species are Anaxyrus, whereas others are Bufo. However, Frost (2022) clearly lists the focal species as Anaxyrus houstonensis:
https://amphibiansoftheworld.amnh.org/Amphibia/Anura/Bufonidae/Anaxyrus/Anaxyrus-houstonensis

Throughout the manuscript, you refer to this species as Bufo houstonensis. If there is a valid reason for doing this, please state it as first mention (ditto Anaxyrus hemiophrys). Otherwise, you have an obligation to follow standard taxonomic nomenclature for this species as you have with others listed.

Other points:

You are not obliged to speculate on the impact of occupancy.

With respect to moving Table 2 to an appendix: PeerJ has no requirement that you do this. It is, however, conventional that results are not repeated. The choice is yours.

I think that it would be correct to include the reviewers in your acknowledgments.